# Simvastatin Inhibits *Brucella abortus* Invasion into RAW 264.7 Cells through Suppression of the Mevalonate Pathway and Promotes Host Immunity during Infection in a Mouse Model

**DOI:** 10.3390/ijms23158337

**Published:** 2022-07-28

**Authors:** Trang Thi Nguyen, Heejin Kim, Tran Xuan Ngoc Huy, Wongi Min, Hujang Lee, Alisha Wehdnesday Bernardo Reyes, Johnhwa Lee, Suk Kim

**Affiliations:** 1Institute of Animal Medicine, College of Veterinary Medicine, Gyeongsang National University, Jinju 52828, Korea; nguyentrang29071996@gmail.com (T.T.N.); khjin0704@gnu.ac.kr (H.K.); txn.huy@hutech.edu.vn (T.X.N.H.); wongimin@gnu.ac.kr (W.M.); hujang@gnu.ac.kr (H.L.); 2Institute of Applied Sciences, HUTECH University, 475A Dien Bien Phu St., Ward 25, Binh Thanh District, Ho Chi Minh City 72300, Vietnam; 3Department of Veterinary Paraclinical Sciences, College of Veterinary Medicine, University of the Philippines Los Baños, Los Baños 4031, Philippines; abreyes4@up.edu.ph; 4College of Veterinary Medicine, Chonbuk National University, Iksan 54596, Korea; johnhlee@chonbuk.ac.kr

**Keywords:** *Brucella abortus*, simvastatin, mevalonate pathway, phagocytosis, immune response, RAW 264.7 cell, ICR mouse

## Abstract

Simvastatin is an inhibitor of 3-hydroxy-3-methylglutaryl CoA reductase and has been found to have protective effects against several bacterial infections. In this study, we investigate the effects of simvastatin treatment on RAW 264.7 macrophage cells and ICR mice against *Brucella* (*B.*) *abortus* infections. The invasion assay revealed that simvastatin inhibited the *Brucella* invasion into macrophage cells by blocking the mevalonic pathway. The treatment of simvastatin enhanced the trafficking of Toll-like receptor 4 in membrane lipid raft microdomains, accompanied by the increased phosphorylation of its downstream signaling pathways, including JAK2 and MAPKs, upon =*Brucella* infection. Notably, the suppressive effect of simvastatin treatment on *Brucella* invasion was not dependent on the reduction of cholesterol synthesis but probably on the decline of farnesyl pyrophosphate and geranylgeranyl pyrophosphate synthesis. In addition to a direct brucellacidal ability, simvastatin administration showed increased cytokine TNF-α and differentiation of CD8^+^ T cells, accompanied by reduced bacterial survival in spleens of ICR mice. These data suggested the involvement of the mevalonate pathway in the phagocytosis of *B. abortus* into RAW 264.7 macrophage cells and the regulation of simvastatin on the host immune system against *Brucella* infections. Therefore, simvastatin is a potential candidate for studying alternative therapy against animal brucellosis.

## 1. Introduction

Brucellosis is one of the most common zoonotic diseases and is still neglected in developing countries. Indeed, it negatively affects not only livestock production but also public health. *Brucella* species are the pathogenic agents of brucellosis. The disease was first discovered by Bruce in 1887 on the island of Malta [1]. Among twelve recently recognized *Brucella* species, *Brucella* (*B.*) *abortus*, together with *B. melitensis* and *B. suis* are mainly the causative agents for brucellosis in cattle and are one of the most pathogenic to humans [2]. *Brucella* usually spreads through direct contact with infected birth tissues, broken skin, and can also be transmitted by contaminated objects. In humans, *Brucella* can be transmitted via the consumption of unpasteurized milk, meat, and animal products from infected animals [3]. Clinical manifestations of cattle brucellosis include abortion, death of young ones, stillbirth, male infertility, and orchitis, resulting in severe losses in the dairy industry [4]. The symptom of human brucellosis in the early stage is high and undulating fever. In the chronic phase, brucellosis causes diverse manifestations, including arthritis, orchitis, hepatitis, encephalomyelitis, and endocarditis [5].

Simvastatin, a lipophilic statin, is an inhibitor of the first committed enzyme, 3-hydroxy-3-methylglutaryl-coenzyme A (HMG-CoA) reductase. This enzyme is responsible for catalyzing the conversion of HMG-CoA to mevalonate. Thus, the reduced cholesterol biosynthesis by simvastatin is primarily due to a decrease in mevalonate synthesis, the second step in the mevalonate pathway. Medical treatment with simvastatin is efficacious in patients with hypercholesterolemia or cardiovascular risk [6]. In addition to the cholesterol-lowering effect, simvastatin exerts anti-inflammatory, anti-oxidative, anti-bacterial, and immunomodulatory properties. Due to the pleiotropic effects, statins group in general and simvastatin, in particular, are highly emerging as potential candidates for host-directed therapy against infectious diseases, even for COVID-19 [7,8,9,10].

The mevalonate pathway is a unique source for the biosynthesis of isoprenoids, a diverse and pivotal class of molecules that exhibit various biological functions, including cell growth/differentiation, gene expression, protein glycosylation, signaling transduction, and vitamins and hormones synthesis. The pleiotropic effects of simvastatin mentioned above rely on the inhibition of the mevalonate pathway. It is related to lowering cholesterol levels and inhibiting key isoprenoids such as farnesyl pyrophosphate (FPP) and geranylgeranyl pyrophosphate (GGPP) that are the intermediate products during the mevalonate pathway. FPP and GGPP play an essential role in the post-translational prenylation of numerous proteins, typically small GTP-binding proteins GTPases [11,12]. This modification of protein enhances their cellular activities.

Currently, the treatment of brucellosis is briefly based on the usage of antibiotics such as doxycycline, rifampicin, streptomycin, co-trimoxazole, and gentamicin in monotherapy or combination therapy. However, although this antibiotic-based therapy has been proven effective for both human and animal brucellosis, it remains controversial and inconsistent [13]. Besides, the relapse rate is one of the most critical goals in brucellosis treatment [14]. Hence, using natural products or drugs that mediate host immune responses to treat brucellosis has received considerable attention and has been widely studied. On the other hand, simvastatin is applicable for particular alternative or adjunctive host-directed therapy for infectious diseases caused by intracellular pathogens [7]. Simvastatin treatment has been proven to mediate host protection against *Listeria* (*L.*) *monocytogenes* by preventing *Listeria*-induced phagosomal escape and increasing IL-12p40 and TNF-α coordinately [15]. Furthermore, simvastatin promotes host immune responses by increasing the secretion of IL-1β, IL-12p70, and IL-10, activating apoptosis and autophagy, resulting in the control of *Mycobacterium* (*M.*) *tuberculosis* infections [16]. However, the effect of simvastatin on *B. abortus* infections has not been studied yet. In the present study, we aimed to investigate the effect of simvastatin treatment on *Brucella* invasion into RAW 264.7 cells and *Brucella* survival in ICR mice.

## 2. Results

### 2.1. Simvastatin Inhibits Internalization of B. abortus into RAW 264.7 but Not Intracellular Growth

First, we aimed to determine the cytotoxic effect of simvastatin and mevalonic acid (mevalonate) on RAW 264.7 macrophage cells. After 48 h of incubation with simvastatin, four high concentrations (75, 100, 200, and 500 µg/mL) showed a significant reduction in the viability of the cells while there was no difference at all the remaining lower concentrations compared to the control cells (Figure 1A). On the other hand, mevalonic acid treatment had no cytotoxic effect at all on the examined cells’ concentrations compared to the control cells (Figure 1B). We next investigated whether simvastatin treatment affects the *B. abortus* invasion into RAW 264.7 cells. The result showed that at a non-cytotoxic concentration of 20 µg/mL, simvastatin significantly reduced the number of bacteria at 15 and 25 min post-infection (pi) (Figure 1C). Interestingly, supplementation of exogenous mevalonate completely abolished the simvastatin-mediated reduction in bacterial internalization (Figure 1C). However, this reduction of *B. abortus* was observed in intracellular assay but was not significant (Figure 1D). To rule out the ability of mevalonate to enhance the invasion of *Brucella*, we applied two non-cytotoxic concentrations of mevalonic acid (20 and 40 µg/mL) to evaluate its direct effect on this abolishment. At all examined time points, no differences in bacterial internalization were observed between treated and control cells (Figure 1E). Therefore, these results suggest that simvastatin initially contributed to reduce *B. abortus* uptake in macrophage cells, probably related to the blockage of the mevalonate metabolic pathway.

### 2.2. Effect of Simvastatin on Brucella Invasion Is Independent of TLR4/JAK2/MAPKs Signaling Pathway

TLR4-associated JAK2 and MAPKs activation play an essential role in the uptake of *Brucella* in macrophages [17]. At first, we isolated the membrane lipid raft microdomains to clarify the distribution of TLR4 on lipid raft upon *Brucella* infection in simvastatin-treated cells. The results display an increase in TLR4 expression in lipid raft extracted from treated cells at 15 min pi (Figure 2A). In addition, phosphorylation of ERK1/2, JNK, and JAK2 was increased in simvastatin-treated cells compared to the control cells at 15 and 25 min pi (Figure 2C,D,F,G). In comparison, simvastatin treatment enhanced the phosphorylation of p38α without *Brucella* infection (Figure 2C,E). These results suggested that the significant inhibition of *Brucella* invasion into macrophages is not related to the lower expression of TLR4 in lipid raft as well as its downstream signaling pathway. The elevation in the activation of this signaling pathway was subjected to other unknown molecular mechanisms related to metabolism.

### 2.3. Simvastatin Inhibits the Uptake of B. abortus by RAW 264.7 Due to Suppression of GGPP and FPP Synthesis Instead of Cholesterol

The 3-(4,5-dimethylthiazol-2yl)-2,5-diphenyl-2H-tetrazolium bromide (MTT) assay showed that cholesterol and FPP treatment had no cytotoxic effect on RAW 264.7 cell viability at all tested concentrations (Figure 3A,C). On the other hand, GGPP significantly decreased cell survivability at 48 h post-incubation with 500 µg/mL, while all the tested concentrations did not display cytotoxicity in cells (Figure 3B). Afterward, to investigate whether simvastatin-reduced cholesterol biosynthesis regulated bacterial internalization, we co-treated the cells with simvastatin (20 µg/mL) and cholesterol (20 or 50 µg/mL) at the same time. The supplementation of exogenous cholesterol did not affect the recovery in simvastatin-mediated decreases of the number of invaded bacteria (Figure 3D). By contrast, the supplement of GGPP (80 µg/mL) or FPP (80 µg/mL) at the same time with simvastatin treatment significantly increased the invasion of *Brucella* into RAW 264.7 cells (Figure 3E). This observation indicated that the inhibition in *Brucella* invasion was not dependent on cholesterol synthesis but regulated via intermediate products of the mevalonate pathway that are FPP and GGPP synthesis.

### 2.4. The Bactericidal Effect of Simvastatin on Brucella Survival

The results show no significant difference in the bacterial survival rate in the presence of 20 µg/mL and 100 µg/mL simvastatin compared to control, but the 200 µg/mL concentration reduced the *B. abortus* survival rate at 48 h post-incubation (Figure 4). These results indicate that simvastatin directly affected the survivability of *B. abortus* at 200 µg/mL concentration.

### 2.5. Simvastatin Administration Promotes Host Defense against B. abortus Infection

After obtaining promising results from the in vitro experiments, we next investigated whether simvastatin can affect the bacterial burden in mice. The oral administration of 20 mg/kg/day of simvastatin in mice displayed a significant reduction in bacterial load in the spleen at 14-day pi compared to the control group (Figure 5A). In line with bacterial burden, the total weight of the spleen collected from simvastatin treatment was lower than the control group (Figure 5B). On the other hand, there were no differences in bacterial load and total weight in the liver in all the examined groups (Figure 5C,D). The level of cytokine production in serum has been known to reflect the host immune responses upon *Brucella* infection [3]. Therefore, we measured the production of six cytokines related to *Brucella* infections, including TNF-α, INF-γ, IL-6, IL-12p70, IL-10, and MCP-1. The result showed an increase in TNF-α secretion in the simvastatin-administered group compared to the control group (Figure 5E). The level of other cytokines in the simvastatin-treated group was not significantly different from the control group (data not shown). Furthermore, CD8^+^ T cells are beneficial in controlling *Brucella* infections. In the present study, simvastatin-treated mice prior to *Brucella* infection showed a marked increase in CD8^+^ T cell differentiation (Figure 5F). These data demonstrated the immunomodulatory effects of simvastatin against *B. abortus* infections in ICR mice.

## 3. Discussion

The invasion of *Brucella* into the host cell is the initial step for a successful infection course. Macrophages in particular are a major target for *Brucella* infection and replication [18]. A previous study reported that simvastatin treatment inhibited the invasion of *Staphylococcus* (*S.*) *aureus* into human umbilical vein endothelial cells (HUVEC), a non-professional-phagocytic cell [19]. In another study done by Loike et al. [20], simvastatin treatment decreased Fc receptor-mediated phagocytosis by mouse peritoneal macrophages and cultured human monocytes. In the present study, simvastatin impaired the internalization of *B. abortus* in macrophages, but this inhibition was abrogated by supplementing mevalonate, even though mevalonate alone did not affect the phagocytosis of *B. abortus* (Figure 1C,E). This result demonstrates the effect of simvastatin treatment on *B. abortus* invasion into RAW 264.7 cells due to the blockage of the mevalonate pathway.

*Brucella* internalizes into macrophages via lipid rafts, a cholesterol-enriched membrane domain. This process requires the recruitment of TLR4 into lipid raft to recognize *Brucella* lipopolysaccharide [21]. Cholesterol depletion disrupts lipid raft integrity, leading to a decrease in transient TLR4 trafficking to lipid raft. Furthermore, cholesterol depletion has been proven to inhibit the invasion of various bacteria into the host cells [22,23,24]. However, in this present study, we found that the expression of TLR4 in the fractionated raft in simvastatin-treated macrophages was higher than in the vehicle control-treated macrophages (Figure 2A,B). This interesting result suggests a possible involvement of another cellular signaling pathway. A study by Jia and colleagues [25] showed that simvastatin treatment increased the transcriptional factor sterol regulatory element-binding protein 2 (SREBP2) expression. SREBP2 restores cellular cholesterol when the cells are deprived of cholesterol, resulting in an accumulation of cholesterol, and consequently upregulates TLR4 protein expression [26]. However, further investigations should be conducted to clarify how SREBP2 interacts with TLR4 upon *Brucella* infection.

MAPK pathway is well-known as a downstream of TLR4/JAK2 signaling pathway that plays an essential role in the phagocytosis of *B. abortus* by macrophages. Reduction of JAK2 and MAPKs phosphorylation decreases the internalization of *B. abortus* into macrophages [17]. Interestingly, although *Brucella* was inhibited, simvastatin treatment enhanced phosphorylation of ERK1/2, JNK, and JAK2 proteins in the infected macrophages (Figure 2C–G). The effects of simvastatin on MAPKs phosphorylation are controversial. Benati and associates [27] previously showed that simvastatin induced JNK phosphorylation in macrophages during phagocytosis of IgG-opsonized *S. aureus*, reducing p38, ERK1/2, and IKK phosphorylation. Another study observed the activation of ERK1/2 phosphorylation by simvastatin in bone marrow-derived macrophages after 24 h of treatment [28]. Besides, simvastatin also significantly enhanced p-JAK2 expression in the liver in the progression of non-alcoholic fatty liver disease in rats [29]. On the other hand, the pre-treatment of simvastatin inhibited phosphorylation of JAK2 and ERK1/2 in renal cancer cells upon IL-6 stimulation [30]. These different results indicate the opposite effect of simvastatin in MAPKs and JAK2 signaling. In the context of this study, the activation effect of simvastatin to MAPKs and JAK2 is likely related to increasing TLR4 in lipid raft. Taken together, we conclude that the attenuation effect of simvastatin on the internalization of *B. abortus* in RAW 264.7 cells is independent of TLR4-linked JAK2 and MAPKs signaling.

Aside from simvastatin not reducing TLR4 trafficking to lipid raft, the co-treatment with cholesterol did not recover the invasion of *Brucella* into macrophages which firmly confirms that the inhibition effect of simvastatin on the internalization of *Brucella* was not a consequence of restraining cholesterol. Thus, another mechanism was considered for this inhibitory effect. Mevalonate cascade is responsible for cholesterol biosynthesis and its intermediates, such as GGPP and FPP [31]. GGPP are the substrates for prenylation of small GTPase such as Rho, Rac, and Cdc42, while FPP is a substrate for prenylation of Ras, another GTPase. These GTPases regulate various phagocytic events in macrophages [20,32,33]. It has been proven that the entry of *B. abortus* into the host cell is via Rho GTPase dependent-manner and the inhibition of Rho GTPase prevents bacterial uptake. Meanwhile, the small GTPases of the Rho subfamily, such as Rho, Rac, and Cdc42, were characterized for their roles in regulating the actin cytoskeleton [33,34]. Thereby, we hypothesize here that simvastatin hampers the production of GGPP and FPP, which can affect the prenylation of GTPase, leading to the reduced phagocytosis of *B. abortus*. As expected, the number of *B. abortus* with GGPP- or FPP-treated cells was significantly higher than for only simvastatin-treated but similar to control-treated cells (Figure 3E). These findings are in agreement with the results of a previous study done by Horn et al. [19] and indicate that simvastatin inhibits the invasion of *B. abortus* via the suppression of GGPP and FPP synthesis. This suppression might block post-translational prenylation of GTPase, leading to the impedance of their membrane localization and causing the reduction of the actin dynamic required for bacterial endocytosis [19]. However, further experiments are necessary to confirm the correlation between GGPP and FPP inhibition and actin dynamics during phagocytosis of *Brucella* in macrophages.

Simvastatin has antimicrobial effects on several bacteria such as *S. aureus*, *Escherichia coli,* and *Acinetobacter baumannii* at high concentrations [35]. Our results show that simvastatin at 200 µg/mL significantly inhibits *Brucella* survivability. Bactericidal effects of simvastatin might reduce the proliferation of *Brucella* in a mouse model. As expected, in the acute phase of brucellosis, we found that simvastatin administration reduced the bacterial burden in the spleen of the infected mice (Figure 5A,B). The protective effect of simvastatin has been reported in *S. pneumoniae-* and *M. tuberculosis*-infected mice [7,36]. The potential effect of simvastatin on host defense against pathogens is not only by its antimicrobial property but also by modulation of host immunity. Cytokines play a crucial role in the initiation, maintenance, and modulation of the host immune responses against *B. abortus*. Numerous studies reported that statin treatment exerts alteration in the cytokine level. Simvastatin enhances the secretion of TNF-α in *L. monocytogenes*-infected macrophages, whereas simvastatin exerts down-regulation of pro-inflammatory cytokines in *Helicobacter pylori*-infected macrophages [15,37]. In this present study, simvastatin administration in mice displayed increased serum TNF-α levels (Figure 5E). TNF-α plays a critical role in macrophage activation, apoptosis, and pro-inflammatory cytokine secretion and is required for host bactericidal activities and *Brucella* infection clearance. During infection, *Brucella* outer membrane protein 25 interferes with the production of TNF-α and other pro-inflammatory cytokines to promote its survival and immune evasion [38]. Therefore, the increase in TNF-α serum levels in the presence of simvastatin is suggested to contribute to the attenuation of *B. abortus* burden in mice. Furthermore, a higher percentage of CD8^+^ T cells was also observed in the serum after 13 days of challenge with *B. abortus* compared to the vehicle control group (Figure 5F). In the KRAS mutant tumor model, simvastatin treatment activated the dendritic cell-mediated CD8^+^ T cells immunity [39]. CD8^+^ T cells are responsible for controlling *B. abortus* infections by perforin- and granzyme-dependent cytolytic activity or by Fas-Fas ligand (Fas/FasL) interactions to destroy the infected host cells [40]. Overall, the immunomodulatory effects of simvastatin suggest its positive contribution to host defense against *B. abortus* infections in mice.

## 4. Materials and Methods

### 4.1. Cell Culture and Bacterial Growth Condition

Murine macrophage RAW 264.7 cells (ATCC, TIB-71), (from passage number 20 to passage number 30) were grown at 37 °C in 5% CO_2_ atmosphere in RPMI 1640 medium (Gibco, CA, USA, 11875119) supplemented with 10% (vol/vol) heat-inactivated fetal bovine serum (FBS) (Gibco, 1600-044) with or without 1% of 100× penicillin/streptomycin (Gibco, 15140122). Depending on particular assays, the cells were seeded at different concentrations in 96-well or 6-well cell culture plates in a culture medium containing RPMI plus 10% FBS. The smooth, virulent, wild-type *B. abortus* 544 (ATCC 23448) was cultivated in Brucella broth (BBL BD, CA, USA) at 37 °C with vigorous shaking for two days for cells or mice infection. The number of bacteria was determined by plating serial dilutions onto a Brucella agar plate followed by three days of incubation at 37 °C for colony-forming unit (CFU) counting.

### 4.2. Reagents Preparation

Simvastatin (Sigma, NJ, USA, S6196) was dissolved in dimethyl sulfoxide (DMSO) (Sigma, D2650-5X) to make concentrations of 1 and 2 mg/mL as stock solutions. Mevalonic acid (Sigma, 50838) and cholesterol (Sigma, C4555-1G) were dissolved in deionized water (DW) to make stock concentrations of 10 and 50 mg/mL, respectively. The stock solution was sterilized by filtration using a 0.22 µm-pore-size membrane. FPP (Sigma, F6892) and GGPP (Sigma, G6025) were stored at −20 °C and directly diluted in PBS before use.

### 4.3. Cell Viability Assessment Assay

The MTT assay was performed to determine the non-cytotoxic concentrations of simvastatin, mevalonic acid, cholesterol, GGPP, and FPP. RAW 264.7 cells were sub-cultured at a density of 3 × 10^4^ cells per well in 96-well plates for 24 h prior to treatment. Cells were then pre-treated with different concentrations of simvastatin (1, 10, 20, 25, 50, 75, 100, 200, and 500 µg/mL), mevalonic acid (10, 20, 50, 100, 200, 500, and 1000 µg/mL), cholesterol (10, 20, 50, 100, 150, 200, 500, and 1000 µg/mL), GGPP (5, 10, 20, 40, 80, 160, 320, and 500 µg/mL), and FPP (5, 10, 20, 40, 80, 160, 320, and 500 µg/mL) in different 96-well plates for 48 h. After that, the medium was changed to a new medium containing 5 mg/mL of MTT solution (Sigma, M5655). After 4 h of incubation with MTT, the medium was removed and incubated with 150 µL of DMSO for 15 min. Finally, the absorbance was then measured at a wavelength of 540 nm using a plate reader (Thermo Labsystems Multiskan, Vantaa, Finland).

### 4.4. Bactericidal Assay

Three concentrations of simvastatin (20, 100, and 200 µg/mL) were utilized to determine anti-bacterial activity against *B. abortus* in a 96-well plate. *Brucella* was grown for two days, pelleted by centrifugation, washed in PBS, and diluted in PBS at approximately 10^4^ CFU per well. After that, different concentrations of simvastatin were added to each well with a total volume of 100 µL per well, while PBS containing 0.1% of DMSO was used as vehicle control. The culture plate was then incubated at 37 °C for 0, 4, 8, 24, and 48 h in a CO_2_ incubator. At the indicated time points, 1:100 dilution was done, and an aliquot of 50 µL was plated onto *Brucella* agar for CFU counting.

### 4.5. Bacterial Invasion and Intracellular Growth Assay

RAW 264.7 cells were prepared as mentioned in the cell viability assay in 100 µL of medium without antibiotics. Cells were then pre-treated and/or co-treated with simvastatin (20 µg/mL), mevalonic acid (20 and 40 µg/mL), cholesterol (20 and 50 µg/mL), and GGPP (80 µg/mL) and FPP (80 µg/mL) with respective negative controls. The co-treatment reagents were added at the same time as simvastatin. Afterward, *Brucella* was deposited onto the cells at a multiplicity of infection (MOI) of 50 by centrifuging at 120× *g* for 10 min and incubated for 5, 15, and 25 min at 37 °C in a CO_2_ incubator. At the designated time points, extracellular bacteria were eliminated by incubating for 30 min in 100 µL of a new medium containing 10% FBS and 50 µg/mL of gentamicin (Gibco, 15710-064). The cells were then washed two times using PBS and lysed with DW. The lysate was diluted at 1:200 in DW, and an aliquot of 50 µL was plated onto Brucella agar for CFU counting. Finally, the bacterial count was converted to log10.

For intracellular growth assay, cells were infected for 1 h at 37 °C. After that, cells were washed twice using PBS and incubated on RPMI containing 10% FBS and 50 µL/mL gentamycin in the presence of simvastatin (20 µg/mL) with or without mevalonate (20 µg/mL) for 4, 24, or 48 h pi. At the indicated time points, cells were washed, lysed, and plated the same as described for the invasion assay. Assays were conducted in triplicates.

### 4.6. Lipid Raft Extraction

A detergent-free method was used to isolate lipid raft as previously described with some modifications [41]. In brief, RAW 264.7 cells were subcultured at 1 × 10^6^ cells per well in 6-well plates and incubated overnight. Cells were then pre-treated with 20 µg/mL of simvastatin for 12 h prior to infection. Infection was performed as that of bacterial invasion assay. At 15 min pi, cells were washed with PBS, lysed in base buffer containing protease inhibitor cocktail (PIC) (GenDEPOT, Barker, TX, USA, P3100-005), and centrifuged at 1000× *g* for 10 min at 4 °C. The post-nuclear supernatant (0.84 mL) was mixed with 1.16 mL of 60% Optiprep Density Gradient Medium (Sigma Aldrich, 92339-11-2) to obtain a 35% final concentration of Optiprep. This mixture was then underlaid to form a discontinuous gradient forming by underlaid method with 2 mL of each Optiprep concentration: 0% (base buffer), 20%, 25%, and 30%. The gradients were separated into nine fractions by centrifugation at 52,000× *g* for 3 h at 4 °C using Beckman ultracentrifuge with an SW 41 Ti rotor (Beckman Coulter, Brea, CA, USA). The distribution of resulting proteins was analyzed by Western blot. All procedures were performed on ice.

### 4.7. Western Blot

RAW 264.7 cells were seeded at a density of 1 × 10^6^ cells per well in 6-well plates. Cells were then pre-treated with simvastatin and infected with *B. abortus* using the same procedure mentioned in the bacterial invasion assay. After that, the total cellular protein was extracted using RIPA Lysis, and extraction buffer (Thermo Fisher, IL, USA, 89900) supplemented with PIC and phosphatase inhibitor (GenDEPOT, P3200-005), followed by BCA protein quantification. Equal amounts of protein samples were subjected to SDS-PAGE and then transferred onto the immobilon-P membrane (Millipore, Burlington, MA, USA). The membrane was blocked with a blocking buffer containing 5% of bovine serum albumin (GenDEPOT, A0100-010) in Tris-buffered saline plus 0.1% of Tween 20 (TBS-T) for 30 min at room temperature (RT). The membrane was next incubated overnight at 4 °C with different primary antibodies diluted in blocking buffer including TLR4 (1:200, Santa Cruz, CA, USA, SC293072), phospho-ERK1/2 (1:500, 4377S), pan-ERK1/2 (1:1000, 4695S), phospho-p38α (1:500, 4511S), pan-p38α (1:1000, 9212S), phospho-JNK (1:500, 9251S), pan-JNK (1:1000, 9258S), phospho-JAK2 (1:500, 3776S), pan-JAK2 (1:1000, 3230S), and β-actin (1:2000, 4967S). The last nine antibodies were purchased from Cell signaling company. After binding with primary antibodies, the membrane was washed using TBS-T buffer three times (20 min each), followed by incubation with goat anti-mouse IgG HRP-conjugated secondary antibody (1:2000, Sigma-Aldrich, Darmstadt, Germany, 32160702) and goat anti-rabbit IgG HRP-conjugated secondary antibody (1:2000, Cell signaling, 7074S) at RT for 1 h, and washed again in TBS-T (10 min × 3). Finally, immunolabelling was detected by chemiluminescent substrate (Atto, WSE-7120L) and visualized in Molecular Imager^®^ ChemiDoc™XRS+ system machine (Bio-Rad Laboratories, Carlsbad, CA, USA).

### 4.8. Mice Treatment with Simvastatin and Protection Experiment

Seven-week-old female ICR mice (Samtako, Osan, Korea) were randomly distributed into four groups of five mice each (non-infected; control and simvastatin treated, infected; control and simvastatin treated groups). The mice were housed per cage with food and water ad libitum for one week of acclimation before receiving treatment. Two groups were orally pre-treated with 20 mg/kg/day of simvastatin for three days, while the mixture of DW with 10% DMSO and 10% Tween-20 was used as vehicle control for two other groups. After that, mice in one simvastatin-treated group and one vehicle control-treated group were intraperitoneally infected with 10^5^ CFU of *B. abortus* in 100 µL per animal, followed by 14 days of continuous daily administration of simvastatin or vehicle control. Other groups were intraperitoneally infected using PBS as the non-infected control groups. Afterward, peripheral blood was collected from the tail vein on day 13 pi to evaluate serum cytokine level and CD8^+^ T cell population. Finally, on day 14 pi, all mice were sacrificed, and the spleen and liver were removed and weighed. To examine the *Brucella* growth in the spleen and liver, 0.05 g of each of these organs were homogenized and suspended in 1 mL of PBS. The homogenates were then made at a 1:10 dilution in PBS and plated onto Brucella agar plate. The number of bacteria was subjected to log10 base calculation to determine the protection unit. The animal handling and sacrifice method in this experiment was reviewed and approved by the Animal Ethical Committee of Chonbuk National University (Authorization Number CBNU-2021-037). All animal experiments were conducted in a Biosafety level 3 laboratory.

### 4.9. Serum Cytokine Level and CD8^+^ T Cell Differentiation in Peripheral Blood Measurement

The serum was separated from peripheral blood by centrifuging at 2000 rpm at 4 °C for 10 min. According to the manufacturer’s instruction, 50 µL of serum samples were applied for analysis of TNF-α, INF-γ, IL-6, IL-12p70, IL-10, and MCP-1 using CBA mouse inflammation kit (BD Biosciences, San Jose, CA, USA, 552364). Data were acquired and analyzed using BD FACVerse flow cytometer and FCAP array software (BD Biosciences, San Jose, CA, USA).

One hundred microliters of peripheral blood were collected and subjected to flow cytometry to determine the differentiation of the CD8^+^ T cell population. A mixture of fresh blood and 75 µL of PE-conjugated rat-anti mouse CD8a monoclonal antibody (BD Pharmingen, San Jose, CA, USA, 553032) was added to a falcon tube and incubated for 10 min in the dark at RT. The mixture was then incubated for 10 min in 2 mL of red blood cell lysis buffer (Roche, Mannheim, Germany, 11814389001) and 3 mL of PBS was added followed by centrifugation at 380× *g* for 5 min. The pelleted cells were washed two times with 3 mL of PBS and re-suspended in 0.5 mL PBS. The data were acquired and analyzed using BD FACVerse flow cytometer and BD FACSuite software (BD Biosciences, San Jose, CA, USA).

### 4.10. Statistical Analysis

The data are expressed as the mean ± the standard deviation (SD). Statistical analysis was performed with GraphPad InStat using unpaired Student’s *t*-test. The results with *, *p* < 0.05; **, *p* < 0.01; and ***, *p* < 0.001 were considered significantly different.

## 5. Conclusions

In conclusion, our results provide evidence for the effectiveness of simvastatin in inhibiting *B. abortus* invasion into RAW 264.7 cells and preventing *B. abortus* proliferation in mice. Interestingly, the inhibition effect on the internalization of *Brucella* was not involved in the reduction of cholesterol synthesis but is possibly dependent on GGPP and FPP synthesis and prenylation of GTPase (Figure 6). In addition, the enhancement of TNF-α production and CD8^+^ T cells differentiation in mice treated with simvastatin demonstrated the correlation between its immunomodulatory effect and the reduction of bacterial load in mice. Despite the limitations, our findings revealed a new potential therapy to control brucellosis. Further exploration focusing on the effect of simvastatin and the mevalonate pathway on the intracellular growth of *B. abortus* is necessary to provide a more comprehensive view of the host–*Brucella* interaction.

## Figures and Tables

**Figure 1 ijms-23-08337-f001:**
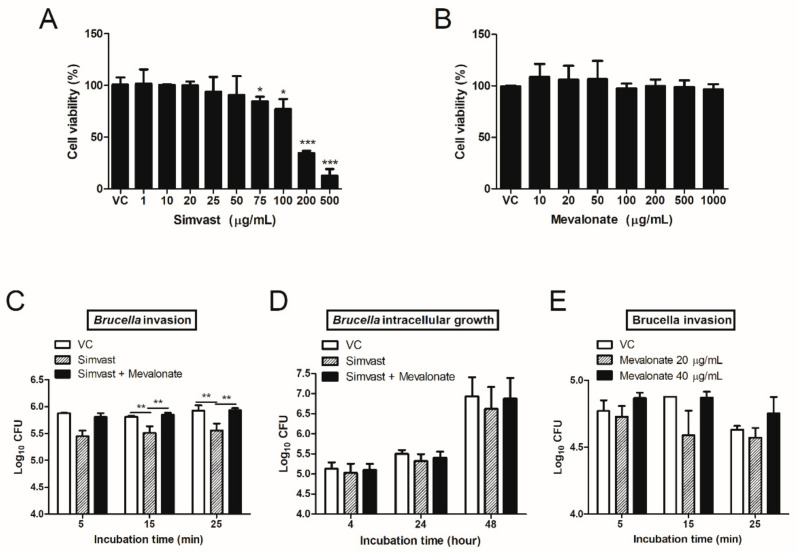
Simvastatin and mevalonic acid treatments affect RAW 264.7 cell viability and *B. abortus* invasion into RAW 264.7 cells. The MTT assay was used to analyze the cytotoxic effects of simvastatin (**A**) and mevalonic acid (**B**) on RAW 264.7 macrophage cells. The *B. abortus* invasion into macrophage cells (**C**) and the *B. abortus* intracellular growth within macrophages (**D**) were evaluated by simvastatin (20 µg/mL) treatment alone or in co-treatment at the same time with mevalonic acid (20 µg/mL). On the other hand, two non-cytotoxic concentrations of mevalonic acid (20 and 40 µg/mL) were applied to determine bacterial internalization individually (**E**). The data are represented as the mean ± SD of duplicate samples from at least two independent experiments. Statistically significant differences relative to control group are indicated by an asterisk (* *p* < 0.05, ** *p* < 0.01, *** *p* < 0.001).

**Figure 2 ijms-23-08337-f002:**
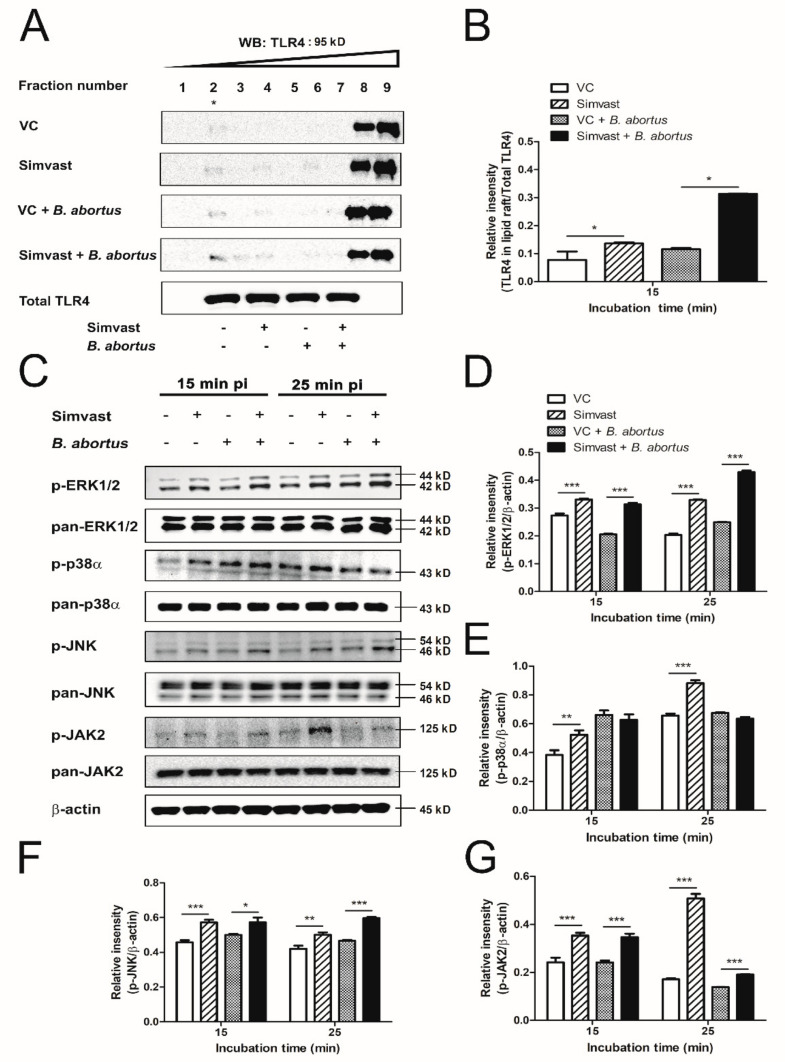
The effect of simvastatin on the activation of signaling pathway upon *B. abortus* infection at 15 and 25 min pi. Cells were pre-treated with 20 µg/mL of simvastatin. At 15 min pi, lipid raft isolation was performed, and the distribution of TLR4 in lipid raft was analyzed using immunoblotting assay (**A**). The lipid raft fractions are indicated by an asterisk in fraction 2, WB: Western blot. ImageJ software was used to calculate the relative intensity of TLR4 in lipid raft compared to total cellular TLR4 (**B**). The downstream signaling pathway of TLR4, including MAPKs (ERK1/2, p38α, and JNK) and JAK2 was also analyzed by immunoblotting assay at 15 and 25 min pi (**C**), and the relative intensity of these proteins normalized to the control β-actin was carried out by ImageJ software (**D**–**G**). The data are represented as the mean ± SD of duplicate samples from at least two independent experiments. Statistically significant differences relative to control group are indicated by an asterisk (* *p* < 0.05, ** *p* < 0.01, *** *p* < 0.001).

**Figure 3 ijms-23-08337-f003:**
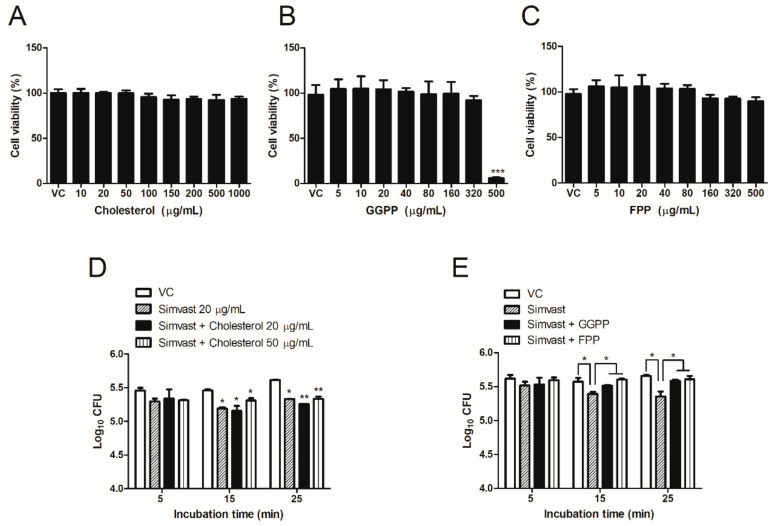
GGPP and FPP, the intermediate products of the mevalonate pathway, affect *Brucella* internalization in RAW 264.7 cells. The MTT assay was utilized to analyze the cytotoxic effects of cholesterol (**A**), GGPP (**B**), and FPP (**C**) on RAW 264.7 macrophage cells. *B. abortus* invasion into macrophage cells was evaluated by co-treatment at the same time with simvastatin (20 µg/mL) and cholesterol (20 and 50 µg/mL) for 12 h prior to infection (**D**). Cells were pre-treated in the presence of simvastatin (20 µg/mL) coordinately either with GGPP (80 µg/mL) or FPP (80 µg/mL) for 12 h prior to infection followed by the performing of bacterial internalization assay (**E**). The data are represented as the mean ± SD of duplicate samples from at least two independent experiments. Statistically significant differences relative to control group are indicated by an asterisk (* *p* < 0.05, ** *p* < 0.01, *** *p* < 0.001).

**Figure 4 ijms-23-08337-f004:**
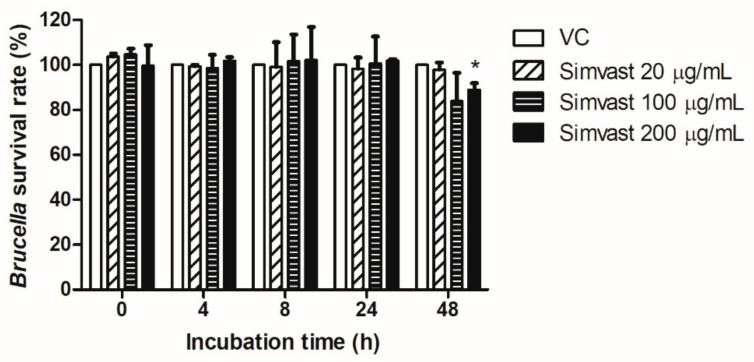
The effect of simvastatin on the direct growth of *B. abortus* was incubated with different concentrations of simvastatin for 0, 4, 8, 24, and 48 h. The data are represented as the mean ± SD of the mean of each group. Asterisks indicate statistically significant differences (* *p* < 0.05).

**Figure 5 ijms-23-08337-f005:**
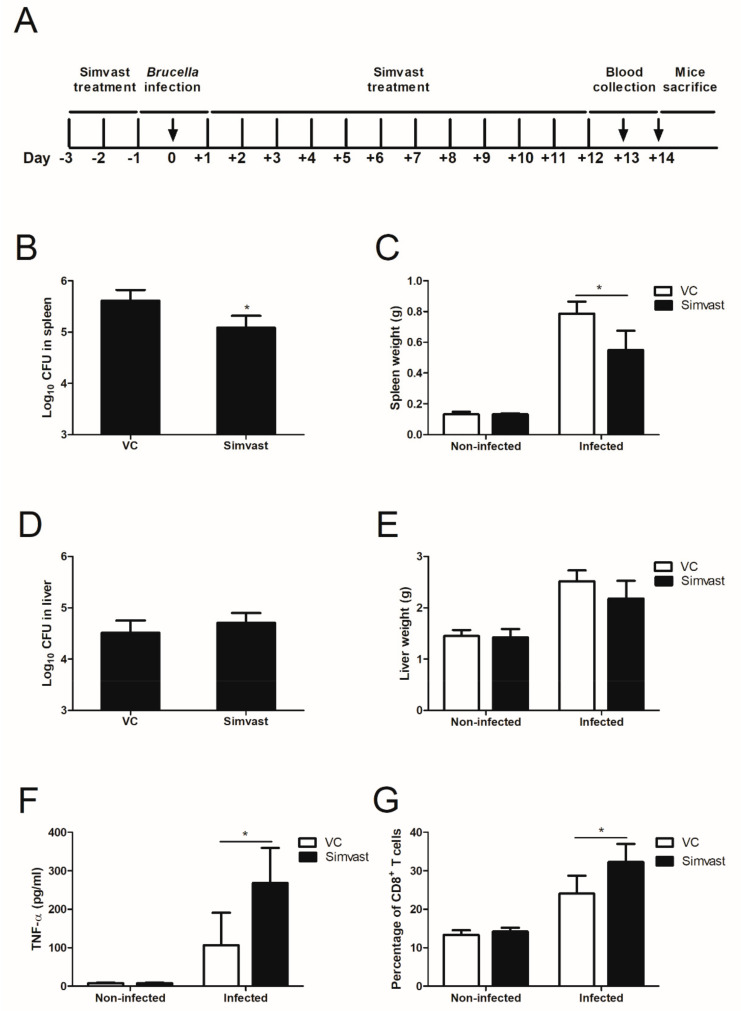
Protection against *B. abortus* in ICR mice treated with simvastatin. ICR mice were orally treated with 20 mg/kg/day of simvastatin or vehicle control three days prior to infection with *B. abortus,* and continuously treated following infection until sacrificed as indicated in the diagram (**A**). At day 14 pi, the bacterial load (**B**) and the total weight of the spleen (**C**) were determined. At the same time, the bacterial burden (**D**) and the total weight of the liver (**E**) were also determined. The production of TNF-α cytokine in serum (**F**) and the differentiation of CD8^+^ T cells (**G**) were analyzed by flow cytometry. The data are represented as the mean ± SD of the mean of each group of five mice. Asterisks indicate statistically significant differences (* *p* < 0.05).

**Figure 6 ijms-23-08337-f006:**
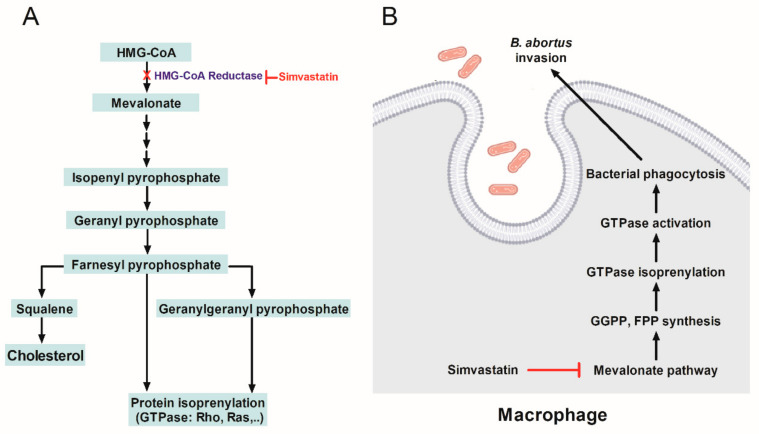
Schematic of mevalonic pathway and illustration of the inhibition effect of simvastatin on *Brucella* invasion through suppression of mevalonate pathway. Simvastatin inhibits the conversion of HMG-CoA to mevalonate, leading to the inhibition of GGPP and FPP synthesis, which regulates the prenylation of some GTPase such as Ras and Rho families (**A**). Inhibiting GTPase synthesis caused by simvastatin treatment affects the activation of GTPase lead to the reduced *B. abortus* internalization into RAW 264.7 cells (**B**). Abbreviations: HMG-CoA: 3-hydroxy-3-methylglutaryl coenzyme A; FPP: farnesyl pyrophosphate; GGPP: geranylgeranyl pyrophosphate.

## Data Availability

Not applicable.

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
