# Peer review of "Simvastatin Inhibits Brucella abortus Invasion into RAW 264.7 Cells through Suppression of the Mevalonate Pathway and Promotes Host Immunity during Infection in a Mouse Model"

_ijms, 2022, doi:10.3390/ijms23158337_

Round 1

Reviewer 1 Report

The manuscript titled Simvastatin, a lipid-lowing agent, inhibits Brucella abortus 544 invasion into RAW 264.7 cells through modulation of mevalonate pathway intermediates and promotes host immunity during infection in a mouse model by authors et al. demonstrated that the simvastatin is a potential candidate for studying alternative therapy against animal brucellosis. In general, this paper seems to be quite interesting and I would like to recommend the acceptance of this work provided that authors can well address the following questions.

1. Compared to the tradditional antibiotics such as doxycycline, rifampicin, etc,

what's the advantage of Simvastatin?

2. The source of brucellosis should be explained clearly in the text ,and the safety level of the laboratory should be described.

Author Response

Author’s Responses to the Reviewer’s Comments

ID: ijms-1803531

Title: Simvastatin inhibits Brucella abortus invasion into RAW 264.7 cells through suppression of the mevalonate pathway and promotes host immunity during infection in a mouse model

Authors: Trang Thi Nguyen, Heejin Kim, Tran Xuan Ngoc Huy, Wongi Min, Hu Jang Lee, Alisha Wehdnesday Bernardo Reyes, John Hwa Lee, Suk Kim

Comments from the reviewers

# Reviewer 1

The manuscript titled simvastatin, a lipid-lowing agent, inhibits Brucella abortus 544 invasion into RAW 264.7 cells through modulation of mevalonate pathway intermediates and promotes host immunity during infection in a mouse model by authors et al. demonstrated that the simvastatin is a potential candidate for studying alternative therapy against animal brucellosis. In general, this paper seems to be quite interesting and I would like to recommend the acceptance of this work provided that authors can well address the following questions.

Q1: Compared to the traditional antibiotics such as doxycycline, rifampicin, etc, what’s the advantage of simvastatin?

Authors answer to Q1: Brucellosis is known as a chronic disease. Using antibiotic treatment such as doxycycline, rifampicin, etc can cause some adverse effects such as hypersensitivity reactions, flu-like syndrome, hematologic effects, hepatotoxicity, gastrointestinal, photosensitivity, superinfection, and tissue hyperpigmentation. Especially, the usage of antibiotics in the long term increases antibiotic resistance and immune cell damage. By contracts, simvastatin enhances the immune system to reduce excessive inflammation, prevents tissue damage, or improves treatment efficacy. Indeed, simvastatin treatment has been proposed to eliminate bacterial infection by modulating the host immune responses. Recently immunotherapy has emerged as a promising alternative therapy for bacterial infection prevention and treatment.

References:

  1. Villate, S. C. A. C., J. C. G., Update of antibiotic therapy of brucellosis. In New insight into brucella infection and foodborne diseases, Ranjbar M, N. M., Mascellino M, Ed. IntechOpen: 2020; pp 1-4.
  2. 16. Guerra-De-Blas, P. D. C.; Bobadilla-Del-Valle, M.; Sada-Ovalle, I.; Estrada-García, I.; Torres-González, P.; López-Saavedra, A.; Guzmán-Beltrán, S.; Ponce-de-León, A.; Sifuentes-Osornio, J., Simvastatin enhances the immune response against Mycobacterium tuberculosis. Frontiers in microbiology 2019, 10, 2097.

Q2: The source of brucellosis should be explained clearly in the text, and the safety level of the laboratory should be described.

Authors answer to Q2: We added the sentenceThe disease was first discovered by Bruce in 1887 on the island of Malta.” in the Introduction section (page 3, lines 45-46). Besides, in this manuscript, we just mentioned the representative Brucella species including Brucella abortus, Brucella melitensis, and Brucella suis, not all of them. Since these species have been considered the most pathogenic to humans.

We also added the sentences “Brucella usually spreads through direct contact with infected birth tissues, broken skin and also can be transmitted by contaminated objects. In humans, Brucella can be transmitted via the consumption of unpasteurized milk, meat, and animal products from infected animals”. In the Introduction section (page 3, lines 48-51).

We also added the sentence “All animal experiments was conducted in a Biosafety level 3 laboratory.” In the Material and Method section (page 17, line 357).

References:

  1. Seleem, M. N.; Boyle, S. M.; Sriranganathan, N., Brucellosis: a re-emerging zoonosis. Veterinary microbiology 2010, 140, (3-4), 392-8.
  2. de Figueiredo, P.; Ficht, T. A.; Rice-Ficht, A.; Rossetti, C. A.; Adams, L. G., Pathogenesis and immunobiology of brucellosis: review of Brucella-host interactions. The American journal of pathology 2015, 185, (6), 1505-17.

Reviewer 2 Report

This manuscript describes the potential use of simvastatin against brucellosis. The presented data shows that simvastatin could inhibit Brucella abortus invasion, CFU count in the spleen and increase inflammatory response. However, the way the manuscript presented is a bit confusing on how simvastatin works and whether it is as effective as current treatment. 

Did the authors perform any histology staining to show that Brucella abortus invasion happens in vivo?

For the in vitro study, it will be good to show some microscopic images that Bruclla abortus invasion happens in RAW264.7 cells and the use of simvastatin can inhibit it.

It will be good to show a schematic diagram on how simvastatin works to inhibit Bruclla abortus invasion.

In vivo study shows that simvastatin sensitizes the inflammatory response cause by Bruclla infection, did the author also show this in vitro?

Any positive control being used for all these experiments?

Fig 1C-E is confusing, it will be good to put some titles to make it more clear what are the differences between these few bar chart

Protein markers are needed for Fig 2

Fig 4, any positive control?

Fig 5. It will be good to provide a schematic diagram on the animal stud design

Fig 5. What is the number of animal used ?

What are the passage number used in this study for RAW264.7 cells

mL instead of ml

Author Response

Author’s Responses to the Reviewer’s Comments

ID: ijms-1803531

Title: Simvastatin inhibits Brucella abortus invasion into RAW 264.7 cells through suppression of the mevalonate pathway and promotes host immunity during infection in a mouse model

Authors: Trang Thi Nguyen, Heejin Kim, Tran Xuan Ngoc Huy, Wongi Min, Hu Jang Lee, Alisha Wehdnesday Bernardo Reyes, John Hwa Lee, Suk Kim

Comments from the reviewers

# Reviewer 2

This manuscript describes the potential use of simvastatin against brucellosis. The presented data shows that simvastatin could inhibit Brucella abortus invasion, CFU count in the spleen and increase inflammatory response. However, the way the manuscript presented is a bit confusing on how simvastatin works and whether it is as effective as the current treatment.

Q1: Did the authors perform any histology staining to show that Brucella abortus invasion happens in vivo?

Authors answer to Q1: We thank the reviewer for conducting an in-depth analysis of our study and providing valuable suggestions. We did not perform histology staining to show the Brucella invasion in mice. We focused on evaluating the bacterial load in the spleen and liver of infected mice after 14 days of infection. The number of CFU in these organs reflects the invasion of Brucella in infected mice. Although the histological study is unfamiliar to us, this comment proposed for us more assay to provide more evidence for the invasion of Brucella in the infected-mice organs, the bacterial burden, and the organ lesions caused by bacterial infection. Therefore, we will perform histological staining in our next study.

Q2: For the in vitro study, it will be good to show some microscopic images that Brucella abortus invasion happens in RAW264.7 cells and the use of simvastatin can inhibit it.

Authors answer to Q2: The microscopic images would be an ideal assay that clearly proves the bacterial invasion happens in RAW 264.7 cells. In our previous data, the bacterial invasion in macrophages showed the same results in microscopic images and plate counting assay. From this reason, we performed the plate counting assay for bacterial invasion. Please understand our situation.

 References:

  1. Reyes , A. W. B.; Arayan, L. T.; Simborio, H. L. T.; Hop, H. T.; Min, W.; Lee, H. J.; Kim, D. H.; Chang, H. H.; Kim S., Dextran sulfate sodium upregulates MAPK signaling for the uptake and subsequent intracellular survival of Brucella abortus in murine Microbial pathogenesis 2016, 91, 68-73.
  2. Reyes , A. W. B.; Hop, H. T.; Arayan, L. T.; Huy T. X. N.; Park, S. J.; Kim, W. D.; Min, W.; Lee, H. J.; Rhee, M. H.; Kwak, Y. S.; Kim S., The host immune enhancing agent Korean red ginseng oil successfully attenuates Brucella abortus infection in a murine model. Journal of ethnopharmacology 2017, 198, 5-14.
  3. Huy T. X. N.; Reyes , A. W. B.; Hop, H. T.; Arayan, L. T.; Son, V. H.; Min, W.; Lee, H. J.; Kim S., Emodin successfully inhibited invasion of Brucella abortus via modulting adherence, microtubule dynamics and ERK signaling pathway in RAW 264.7 cells. Journal of microbiology and biotechnology 2018, 28(10), 1723-9.
  4. Hop, H. T.; Reyes , A. W. B.; Huy T. X. N.; Arayan, L. T.; Min, W.; Lee, H. J.; Rhee, M. H.; Chang H. H.; Kim S., Interleukin 10 suppresses lysosome-mediated killing of Brucella abortus in cultured macrophages. Journal of biological chemystry 2018, 293(9), 3134-44.

Q3: It will be good to show a schematic diagram on how simvastatin works to inhibit Brucella abortus invasion.

Authors answer to Q3: As the reviewer suggested, we added Figure 6 to show how simvastatin inhibits Brucella abortus invasion into RAW 264.7 cells. We also added the Figure 6 legend in the Figure legends section (page 27, lines 588-594).

Q4: In vivo study shows that simvastatin sensitizes the inflammatory response cause by Bruclla infection, did the author also show this in vitro?

Authors answer to Q4: In in vitro experiment, we checked the effect of simvastatin on the intracellular growth of Brucella, but there were no significantly differences in the number of intracellular Brucella CFU (Figure 1D; and in the Results section, page 5, lines 103-104). Therefore, the level of cytokine secreted by infected-RAW 264.7 cells were did not check in the context of this study. Even though the effect of simvastatin on intracellular growth of B. abortus is not significant, we still noticed that “Further exploration focusing on the effect of simvastatin and the mevalonate pathway on the intracellular growth of B. abortus is necessary to provide a more comprehensive view of the host-Brucella interaction” in the Conclusion section (page 18, lines 386-389). The measurement of cytokine secretion to investigate the inflammatory response is one of the important factors that decide the intracellular growth of Brucella. We have to proceed with this cytokine measurement in any experiments which show a significant difference in intracellular growth of Brucella.

Q5: Any positive control being used for all these experiments?

Authors answer to Q5: In this paper, we only used vehicle control for all our experiments. We did not use the positive control. Reviewer 3 also mentioned the positive control for MTT assay. It should include such kinds of positive control in biological experiments. We will include the positive control in our further experiments.

Q6: Fig 1C-E is confusing, it will be good to put some titles to make it more clear what are the differences between these few bar chart

Authors answer to Q6: We added titles to the Figures 1C, 1D, and 1E. However, Figures 1C and 1E are the same assays with different purposes, so we added the same title for these figures. The differences between these two figures were described in the Figure legends section (page 25, lines 545-548) and we also addressed the reason why we experimented with Figure 1E in the results section (page 5, lines 104-106).

Q7: Protein markers are needed for Fig 2

Authors answer to Q7: As per reviewer’s suggestion, we added markers for protein size for Figure 2.

Q8: Fig 5. It will be good to provide a schematic diagram on the animal stud design

Fig 5. What is the number of animal used?

Authors answer to Q8: As per reviewer’s suggestion, we added a schematic diagram (Figure 5A) to show the animal study design. We also added the description for Figure 5A in the Figure legend section (page 26, lines 579-582).

As mentioned in the materials and methods, we carried out this in vivo experiment with four groups; each group has five mice. This experiment was repeated twice, so we used a total of 40 mice for the in vivo experiment. In the Material and Methods section (page 16, lines 341-343).

Q9: What are the passage number used in this study for RAW264.7 cells, mL instead of ml.

Authors answer to Q9: The passage number of cells we used in this study is from passage number 20 to passage number 30. We added information about passage number of cells to the sentence “Murine macrophage RAW 264.7 cells (ATCC, TIB-71), (from passage number 20 to passage number 30) were grown at 37oC in 5% CO2 atmosphere in RPMI 1640 medium (Gibco, 11875119) supplemented with 10% (vol/vol) heat-inactivated fetal bovine serum (FBS) (Gibco, 1600-044) with or without 1% of 100x penicillin/streptomycin (Gibco, 15140122).” In the Materials and methods section (page 12, lines 250-253).

Following the suggestion of the peer reviewer, we replaced all “ml” by “mL” in the manuscript and all figures.

Reviewer 3 Report

The manuscript “Simvastatin, a lipid-lowing agent, inhibits Brucella abortus 544 invasion into RAW 264.7 1 cells through modulation of mevalonate pathway intermediates and promotes host immunity 2 during infection in a mouse model” by Nguyen et al. is well written, presents a well-structured design and rationale, appropriate methodology and interesting results. The discussion is well developed and supported by the literature, focusing on all the results obtained. Thus, I suggest only small changes and corrections, namely:

Title

- the title seems too long, so I suggest shortening it.

Abstract

- line 28, delete "by using the immunoblotting assay" because it is not a relevant information and is misspelled in this sentence.

- line 44, change the verb "affected" to the present tense (affects), since it is still a current problem.

- include the clinical manifestations of brucellosis in humans, as it also applies.

- line 59, delete "the current most dangerous disease in the world" and leave only COVID-19, because it is a controversial statement.

Results

- line 102, include "reduce": “these results suggest that simvastatin initially contributed to reduce B. abortus uptake in macrophage cells”

- change title 2.2 to "effect of simvastatin on Brucella invasion..." since "simvastatin-mediated" suggests that simvastatin has a direct molecular role in Brucella uptake.

- line 107, the reference does not support the assertion. Add the correct reference.

- line 113, rewrite the sentence, since “sinvastatin inhibits Brucella invasion” is not a conclusion of the results of this section: "These results suggest that the significant inhibition of Brucella invasion into macrophages is not related to the lower expression of TLR4 in lipids raft as well as its downstream signaling pathways".

- The MTT must have a positive control included in order to demonstrate that the lack of cytotoxicity is not due to a problem in the assay. If the authors did not use any, then, since this fault does not jeopardize further results, these results can be accepted only in order not to entail repeating all MTT assays. However, in the future, it will be a situation to correct.

- in graphs B-G of figure 2, include the y-axis unit.

- in the legend of figure 2, correct "The effect of simvastatin on the signaling pathway..." to "the effect of simvastatin on the activation of the TLR4/JAK2/MAPK signaling pathway..."

- in the legend of figure 4, line 566, a period and “B. abortus” are missing.

Author Response

Author’s Responses to the Reviewer’s Comments

ID: ijms-1803531

Title: Simvastatin inhibits Brucella abortus invasion into RAW 264.7 cells through suppression of the mevalonate pathway and promotes host immunity during infection in a mouse model

Authors: Trang Thi Nguyen, Heejin Kim, Tran Xuan Ngoc Huy, Wongi Min, Hu Jang Lee, Alisha Wehdnesday Bernardo Reyes, John Hwa Lee, Suk Kim

Comments from the reviewers

# Reviewer 3

The manuscript “Simvastatin, a lipid-lowing agent, inhibits Brucella abortus 544 invasion into RAW 264.7 1 cells through modulation of mevalonate pathway intermediates and promotes host immunity during infection in a mouse model” by Nguyen et al. is well written, presents a well-structured design and rationale, appropriate methodology and interesting results. The discussion is well developed and supported by the literature, focusing on all the results obtained. Thus, I suggest only small changes and corrections, namely:

Q1:  Title: the title seems too long, so I suggest shortening it.

Authors answer to Q1: As the suggestion of the reviewer, we changed the title toSimvastatin inhibits Brucella abortus invasion into RAW 264.7 cells through suppression of the mevalonate pathway and promotes host immunity during infection in a mouse model”.

Q2: Abstract

Q2.1:  Line 28, delete “by using the immunoblotting assay” because it is not a relevant information and is misspelled in this sentence.

Authors answer to Q2.1: As per reviewer’s suggestion, we changed the sentence “Treatment of simvastatin enhanced the trafficking of Toll-like receptor 4 in membrane lipid raft microdomains, accompanied by increased phosphorylation of its downstream signaling pathways, including JAK2 and MAPKs, upon Brucella infection by using the immunoblotting assay” by “Treatment of simvastatin enhanced the trafficking of Toll-like receptor 4 in membrane lipid raft microdomains, accompanied by increased phosphorylation of its downstream signaling pathways, including JAK2 and MAPKs, upon Brucella infection” in the Abstract section (page 2, lines 25-28).

Q2.2:  Line 44, change the verb “affected” to the present tense (affects), since it is still a current problem.

Authors answer to Q2.2: We changed the sentence “Indeed, it negatively affected not only livestock production but also public health.” by “Indeed, it negatively affects not only livestock production but also public health.” in the Introduction section (page 3, line 44, 45).

Q2.3:  Include the clinical of brucellosis in humans, as it also applies.

Authors answer to Q2.3: As the suggestion of the reviewer, we added the sentence “The symptom of human brucellosis in the early stage is high and undulating fever. In the chronic phase, brucellosis causes diverse manifestations, including arthritis, orchitis, hepatitis, encephalomyelitis, and endocarditis.” In the introduction section (page 3 line 53-55). We added a new reference in the References section (page 20, lines 420-422).

References:

  1. Dean, A. S.; Crump, L.; Greter, H.; Hattendorf, J.; Schelling, E.; Zinsstag, J., Clinical manifestations of human brucellosis: a systematic review and meta-analysis. PLoS neglected tropical diseases 2012, 6, (12), e1929.

Q2.4:  Line 59, delete “the current most dangerous disease in the world” and leave only COVID-19, because it is a controversial statement.

Authors answer to Q2.4: As the reviewer suggested, we changed the sentences “Due to pleiotropic effects, statins group in general and simvastatin, in particular, are highly emerging as potential candidates for host-directed therapy against infectious diseases, even for the current most dangerous disease in the world COVID-19.” to “Due to pleiotropic effects, statins group in general and simvastatin, in particular, are highly emerging as potential candidates for host-directed therapy against infectious diseases, even for COVID-19.” in the introduction section (pages 3, 4 lines  63-55)

Q3: Results

Q3.1:  Line 102, include “reduce”: “these results suggest that simvastatin initially contributed to reduce B. abortus uptake in macrophage cells”

Authors answer to Q3.1 As per reviewer’s suggestion, we changed the sentences “Therefore, these results suggest that simvastatin initially contributed to B. abortus uptake in macrophage cells, probably related to the blockage of the mevalonate metabolic pathway.” to “Therefore, these results suggest that simvastatin initially contributed to reduce B. abortus uptake in macrophage cells, probably related to the blockage of the mevalonate metabolic pathway.” in the Results section (pages 5, 6  lines 107-109).

Q3.2: Change title 2.2 to “effect of simvastatin on Brucella invasion...” since “simvastatin-mediated” suggests that simvastatin has a direct molecular role in Brucella uptake.

Authors answer to Q3.2: We changed the title 2.2 “Simvastatin-mediated Brucella invasion is independent on TLR4/JAK2/MAPKs signaling pathway” to “Effect of simvastatin on Brucella invasion is independent on TLR4/JAK2/MAPKs signaling pathway” in the Results section (page 6, line 110, 111)

Q3.3:  Line 107, the reference does not support the assertion. Add the correct reference.

Authors answer to Q3.3: We changed the reference for the sentence: “TLR4-associated JAK2 and MAPKs activation play an essential role in the uptake of Brucella in macrophages [17]” to the correct reference in the References section (page 21, lines 451-454)

The correct reference:

  1. 17. Lee, J. J.; Kim, D. H.; Kim, D. G.; Lee, H. J.; Min, W.; Rhee, M. H.; Cho, J. Y.; Watarai, M.; Kim, S., Toll-like receptor 4-linked Janus kinase 2 signaling contributes to internalization of Brucella abortus by macrophages. Infection and immunity 2013, 81, (7), 2448-58.

Q3.4:   Line 113, rewrite the sentence, since “simvastatin inhibits Brucella invasion” is not a conclusion of the results of this section: “These results suggest that the significant inhibition of Brucella invasion into macrophages is not related to the lower expression of TLR4 in lipids raft as well as its downstream signaling pathways”.

Authors answer to Q3.4: As the reviewer suggested, we replaced the sentence “These results suggested that simvastatin significantly inhibited Brucella invasion into macrophages, but this event was not dependent on the expression of TLR4 in lipid raft as well as its downstream signaling pathway.” by the sentence “These results suggested that significant inhibition of Brucella invasion into macrophages is not related to the lower expression of TLR4 in lipid raft as well as its downstream signaling pathway.” in the results section (page 6, lines 119-121).

Q3.5:   The MTT must have a positive control included in order to demonstrate that the lack of cytotoxicity is not due to a problem in the assay. If the authors did not use any, then, since this fault does not jeopardize further results, these results can be accepted only in order not to entail repeating all MTT assays. However, in the future, it will be a situation to correct.

Authors answer to Q3.5: We agree that positive control should be included in order to demonstrate that the lack of cytotoxicity is not due to a problem in the assay. In the current study, to avoid the problem in the assay, MTT assay were repeated three independent replicates. Moreover, in each replicate, we have triplicated wells for each concentration. However, we will supplement the positive control for MTT assay in further.

Q3.6:  In graphs B-G of figure 2, include the y-axis unit.

Authors answer to Q3.6:  We thank the reviewer for giving a worth considering suggestion. However, in this experiment, we normalized the western blot results based on the intensity of the interested proteins. This y-axis in graphs B, D-G shows the relative intensity of phosphorylated proteins, which is calculated by the ratio of the intensity phosphorylated protein bands per the intensity of corresponding β-actin protein band. Therefore, we do not have any particular units for this kind of ratio. We understand that the unit here is the intensity ratio between proteins of interest and the control gene β-actin.

Q3.7:  In the legend of figure 2, correct “The effect of simvastatin on the signaling pathway...” to “the effect of simvastatin on the activation of the TLR4/JAK2/MAPK signaling pathway...”

Authors answer to Q3.7: As the reviewer suggested, we replaced the sentence “The effect of simvastatin on the signaling pathway upon B. abortus infection at 15 and 25 min pi.” by “The effect of simvastatin on the activation of signaling pathway upon B. abortus infection at 15 and 25 min pi.” in the Figure legends (page 26, lines 553-554)

Q3.8:  In the legend of figure 4, line 566, a period and “B. abortus” are missing.

Authors answer to Q3.8: We changed the word “B abortus” to “B. abortus” in the Figure legends section (page 27, lines 575).

Round 2

Reviewer 2 Report

Author addressed most of my comments